# Moral Relevance Approach for AI Ethics

Shuaishuai Fang 

School of Philosophy, Shanxi University, Taiyuan 030006, China; 201910602002@email.sxu.edu.cn

**Abstract:** Artificial intelligence (AI) ethics is proposed as an emerging and interdisciplinary field concerned with addressing the ethical issues of AI, such as the issue of moral decision-making. The conflict between our intuitive moral judgments constitutes an inevitable obstacle to decision-making in AI ethics. This article outlines the Moral Relevance Approach, which could provide a considerable moral foundation for AI ethics. Taking moral relevance as the precondition of the consequentialist principles, the Moral Relevance Approach aims to plausibly consider individual moral claims. It is not only the common ethical target shaping our moral consensus but also the inherent moral ability connecting others with us.

**Keywords:** AI ethics; moral relevance; consequentialism; contractualism; separateness

## 1. Introduction

With the significant enhancements in artificial intelligence (AI) technology, a great number of ethical concerns arise. The widespread application of AI technology inevitably has a considerable impact on the existing social order and human ethical cognition. AI ethics is thus proposed as an emerging and interdisciplinary field concerned with addressing the ethical issues of AI. What AI ethics is most concerned with is the ethical risks brought by AI systems, such as privacy leakage, discrimination, unemployment, and morally wrong decision-making. Regarding moral decision-making, AI technology has been gradually implemented to tackle various problems related to ethical decision-making, such as those encountered by autonomous vehicles [1] (p. 630) and in image-based diagnostics [2] (p. 163). Some AI systems attempt to integrate certain moral normative principles into their algorithms. For instance, Derek Leben presented a way of developing Rawls' Contractarian moral theory into an algorithm for car crash optimization in autonomous vehicles [3] (p. 114). According to this approach, AI could decide who should be saved first and which decision is morally right using given normative principles, such as the Maximin principle. Compared with Utilitarianism, Kantian Ethics, Virtue Ethics, and Prima Facie approaches, Derek Leben posits that Rawlsian Contractarianism could be more appropriate. Faced with the contradictory moral claims of particular moral situations, Derek Leben chooses the Maximin principle as the normative principle of AI ethics. The problem lies in the fact that Rawlsian Contractarianism cannot integrate our human consequentialist and nonconsequentialist ideas satisfactorily. What Rawlsian Contractarianism tries to provide is a considerable ethical principle for moral decision-making. The major risk of AI moral decision-making is not only the uncertainty caused by the opacity of technology, but also that its astonishingly rapid growth has exceeded the boundaries of human ethical cognition. What hinders AI ethics is that people hold different normative claims due to their diverse ethical concerns.

This article outlines the Moral Relevance Approach for AI ethics, which could provide a considerable moral foundation for connecting others with us. The Moral Relevance Approach is proposed to integrate our contradictory intuitive moral judgments, especially in moral situations that both consequentialism and nonconsequentialism cannot satisfactorily explain.

The distinctiveness of AI ethics is introduced first. A concrete normative principle is needed in moral decision-making in AI ethics. Meanwhile, the ordinary moral intuitive claims that people hold to make moral judgments are in conflict with each other. Bridging abstract ethical principles with contradictory intuitive claims constitutes the major distinctiveness of AI ethics. What the Moral Relevance Approach supplies is the strength of moral relevance that should be taken as the necessary precondition of the utilitarian principle. Whether the action is morally relevant to the moral agents is a primary distinction that cannot be ignored. The concise reasons why the Moral Relevance Approach should be accepted are clarified herein.

To satisfy the concrete practical requirements of AI ethics, a unique ethical foundation is provided by the Moral Relevance Approach to integrate our human contradictory intuitive judgments. What the Moral Relevance Approach adopts could be taken as a normative approach, which aims to provide a corresponding and reasonable explanation of our different moral judgments. This means that the decisions made by AI should be consistent with human moral judgments. According to the Moral Relevance Approach, an action is morally wrong because it is excluded by some set of moral normative principles that people cannot reasonably reject. This yields to the redundancy objection, and further justification for the necessity of the reasonable rejection of the Moral Relevance Approach is given in this paper.

## 2. Distinctiveness of AI Ethics
*Consider a Case of AI Ethics*

Risky Highway: An autonomous vehicle is driving on the highway, and an unexpected accident occurs. The AI of this vehicle could save five passersby and let the driver die, or it could save the driver and let the five passersby die. The AI can only make one single choice at this time. Choosing to save the driver means that the five passersby will be bound to die, and vice versa.

If this autonomous vehicle takes consequentialism as a guide for its actions, it is easy to determine that five deaths are obviously worse than one death. Consequentialism holds that moral normative properties depend only on consequences, which means that the vehicle should choose the behavior that will lead to better outcomes. Following this interpretation, letting the driver die and saving five passersby will be the only morally right choice. On the other hand, if this autonomous vehicle behaves on the basis of nonconsequentialism, saving the driver will also be reasonably morally right. As recent research shows, people who are a third party are inclined toward consequentialism, but they dislike it when they are in the driver's position [4] (p. 1573). This leads to the following question: What moral norms should be implemented in the AI of a self-driving vehicle? What hinders us is the conflict between consequentialism and nonconsequentialism. To illuminate the issue of moral decision-making in AI ethics, there are two distinctions that must be clarified, and these are outlined below.

First, specific practical requirements arise in AI ethics. It is easy to regard Risky Highway as a modified version of the trolley problem. The trolley problem is also intriguing due to the conflict between consequentialism and nonconsequentialism. In these cases, all of the alternative choices can find support and justification in their corresponding moral normative theories. However, there is an essential difference that needs to be clarified. The trolley problem was proposed to test the difference between the acceptable and intended consequences of an action. The trolley problem does not represent any actual ethical situation; this type of thought experiment should not be resolved by the so-called morally right or wrong judgment. According to Kamm, such problems are designed to investigate the difference between ethical ordinary intuitions and normative theories [5] (pp. 15–16).

On the contrary, the AI is forced to respond immediately during the same ethical situations. Various applications of AI, such as smartphones, autonomous vehicles, medical images, and autonomous weapons, put forward more specific practical requirements for AI ethics. As we observed in Risky Highway, there is a unique requirement for AI to

immediately respond to moral situations. Moreover, the ethical decision-making in the trolley problem was constructed in ideal hypothetical conditions. The participants are taken as full-blown moral agents with full intelligence and complete ethical normative knowledge, and they act without any subjective prejudices. Nevertheless, these hypothetical conditions seem to be too ideal for AI ethics to achieve. To make the AI more transparent, a concrete normative principle about how it will act is required. Unlike in the philosophical investigation in the trolley problem, a concrete moral decision must be given in this type of Risky Highway scenario.

Second, the ordinary moral intuitions that people rely on in moral decision-making are in conflict with each other. AI ethics not only appeal to a considerable normative principle but also are confronted with the conflict between different moral intuitions. Just as we found in Risky Highway, consequentialism and nonconsequentialism could both apply to moral decision-making, and justifications could be established for the corresponding ethical judgments. What consequentialism pursues is the greatest level of happiness for the highest number of people. Thus, saving five passersby and causing one death corresponds to the consequentialist normative principle, while nonconsequentialism holds that integrated benefit is not the only criterion of moral judgment. Even when the moral interest of greater numbers is at stake, an individual's moral claim can also be ethically right. The AI is liable to save its owner—in this case, the driver—from possible dangers.

Which normative theory should AI adopt? Considering the conflict between consequentialism and nonconsequentialism, it seems difficult to decide which one is better. "The task of moral philosophy is largely one of seeking a coherent, systematic, and non—ad hoc way of accommodating and explaining many of our pretheoretical intuitive judgments as well as most of our firm, considered, ones" [6] (p. 7). However, the crucial question is not regarding the normative approaches themselves, but whether the moral intuition and considered judgments can coexist with each other. Both consequentialism and nonconsequentialism are faced with the same problems in AI ethics. The contradictory intuitive ethical judgments turn out to be the major distinctiveness of AI ethics, and bridging abstract ethical principles with specific moral situations becomes the most urgent problem that AI needs to solve.

### 3. Moral Relevance Approach

Neither consequentialism nor nonconsequentialism can effectively resolve the conflict in our intuitive moral judgments in the AI ethical case, as we just saw. The AI appeals to a considerable moral foundation that could reasonably integrate our different and contradictory normative judgments. This is where the Moral Relevance Approach comes in. This approach could provide a coherent moral basis for our contradictory intuitions and judgments. Below, we consider a variation of the Risky Highway scenario.

Risky Electricity: An autonomous vehicle is driving along the highway, and an unexpected situation occurs. The AI of this vehicle has to make an immediate moral decision: saving the driver or allowing a nearby city to experience an insignificant power failure for a short time, which will influence thousands of people. Compared with losing a life, a short period of inconvenience seems more acceptable to most of us.

It seems tenable that we should save the driver's life instead of preventing the power failure. Although avoiding the power failure represents a greater combined benefit, saving the driver is still the morally right choice to make. As Johann Frick puts it, "Ethical theories, like classical utilitarianism, that defend interpersonal aggregation hold that in evaluating an action, we should sum the benefits and losses it imposes on different people to obtain an aggregate quantity" [7] (p. 175). The problem lies in the fact that sticking to the pursuit of combined benefits will result in counterintuitive moral judgments being made that we wish to avoid. Consequentialism is not the most appropriate moral normative principle in this type of scenario. Because losing a life is more serious than the inconvenience of a short power failure for most of us, protecting the integrated benefits from possible harm would

not take moral priority in the Risky Electricity scenario. It is clear that the AI has a more convincing moral reason to save the individual than to protect the integrated benefits.

Moral correctness does not depend on the number of people who will be influenced at a given moment. Finitely weighted total prioritarianism, which is considered a promising alternative version of utilitarianism, is faced with the same questionable property: "it justifies that a significant benefit to the worst-off person can be outweighed by trivial benefits to enough best-off people" [8] (p. 258). The strength of moral seriousness should be introduced in moral decision-making. Here, the Moral Relevance Approach can be considered:

Before applying the utilitarian principle, it is necessary to confirm whether the decision-making is morally relevant to the person involved.

The key distinction is not the number of integrated benefits but the moral seriousness of the individual's moral claim. In Risky Electricity, losing a life is morally relevant, while a power cut is not as morally relevant. We should make a choice according to which is morally relevant but not good for the quantity of integrated benefits. According to Scanlon, if "aggregative arguments are not appropriate, then it seems that our intuitive moral thinking is best understood in terms of a relation of 'relevance' between harms" [9] (p. 239). That is to say, before applying the utilitarian principle, whether a decision is morally relevant to the people who are influenced must be clarified first.

What the Moral Relevance Approach applies is still contractualist in nature. It yields the same ethical decision: saving the driver's life instead of preventing a trivial and irrelevant power failure in Risky Electricity. Here, the burden of death exceeds the inconvenience of power failure. Consequently, we should abandon the consequentialist principle and give concern to the individual moral claim in certain cases like the Risky Electricity scenario. Moreover, as Iwao Hirose puts it, "aggregation skeptics must commit themselves to the counterintuitive claim that the numbers of individuals who would be affected by a certain action makes no moral difference" [10] (p. 1). The aggregated approach adopted by consequentialism will cause latent counterintuitive implications that we must try to avoid.

There is a shift in characterizing the distribution of benefits and burdens of moral agents and people who will be influenced. What consequentialism adopts is an ex post view. The ultimate gain and loss after the moral decision-making are what consequentialism is concerned with, while nonconsequentialism focuses more on whether the individual moral claims have been fully respected ex ante. It gives greater consideration to the motivations and reasons people hold ex ante in moral decision-making. According to Scanlon, "This might be justified on an ex ante basis because, if the number of beneficiaries is sufficiently great, and if we have no reason ex ante to believe that one is more likely to be the victim in a situation of this kind than to be a beneficiary, then one has no reason, ex ante to object to this principle" [11] (p. 510). Different points of view decide their different theoretical goals. What consequentialism concerns is which ex post outcome is better for us, while nonconsequentialism focuses on who should be saved ex ante.

Being neither consequentialism nor nonconsequentialism, the Moral Relevance Approach aims to integrate our contradictory intuitive judgments in cases like Risky Highway and Risky Electricity. Individual moral claims possess a critical position in moral decision-making, which should not be ignored. By introducing moral relevance as the precondition of the utilitarian principle, the Moral Relevance Approach could offer a considerable interpretation of our intuitive moral judgments. The Moral Relevance Approach supplies a normative framework that could satisfactorily integrate our intuitive judgments in line with consequentialism and nonconsequentialism. The upshot is that consequentialism and nonconsequentialism cannot satisfactorily account for why we have contrary normative judgments in AI cases. The combined benefits seem to not be the constant behavioral principle that could be applied to specific situations. What the Moral Relevance Approach provides is a considerable reason and justification for us to restrain such a consequentialist principle.

## 4. Reasons for the Moral Relevance Approach

The crucial step of moral decision-making in AI ethics is to identify a transparent way to explain how our contradictory judgments behave in moral cases like Risky Highway and Risky Electricity. Consequentialism and nonconsequentialism adopt conflicting ways of treating individual moral value. Consequentialism takes individual moral value as a fixed and limited object that can be calculated using the utilitarian principle. Following this, the moral agent is viewed as an isolated and indifferent carrier of ethical values. As a consequence, individual moral claims should have no place in moral decision-making, since moral values do not vary from moral agent to moral agent. However, as we found in Risky Electricity, the attitudes and intentions of the moral agent could also generate a significant impact on our moral choices. For the people who are influenced, the quantity is irrelevant to the final moral decision. It becomes essential to introduce the morally relevant factors of agents into the moral decision-making process. What follows is a concise sketch of the moral reasons why we should take moral relevance as a constraint on the utilitarian principle.

First, the Moral Relevance Approach does not establish moral rightness by merely relying on the ex post combined benefits. Consequentialism takes the combined benefits as the highest normative criterion and, thus, cannot distinguish the different behaviors or affairs with the same consequence. For example, killing someone and letting someone die are morally equivalent per se, according to consequentialist principles, because they have the same expected outcome: someone's death. It is hard to determine the moral differences between killing someone and letting someone die, according to consequentialism, despite them being obviously distinct behaviors with completely different convictions in common-sense morality. According to F. M. Kamm, killing someone and letting someone die have distinct ethical evaluations due to the difference in their moral agents' diverse motivations and intentions [12] (p. 18). In killing someone, the moral agent has intentionally initiated an interference with the victim, while in letting someone die, the moral agent fails to act, thus avoiding any kind of interference with possible danger. Distinguishing between these two behaviors appeals to a viable way to reasonably integrate the morally relevant claims of the moral agents involved. This instance is presented to illustrate how AI cannot make moral judgments if it merely relies on the consequentialist principle, since different behaviors with the same consequence have different moral evaluations due to their various motivations and intentions. Thus, it is essential to introduce moral relevance into the moral decision-making of AI ethics.

Second, the Moral Relevance Approach provides an accurate and explicit explanation, especially in moral situations like Risky Electricity. Compared with consequentialism, the Moral Relevance Approach does not make a moral judgment based entirely on the integrated benefits that behaviors may generate. Completely endorsing consequentialism will cause the issue to become even more confusing. The normative principle that consequentialism adopts turns out to be not as convincing as suggested. Scanlon offers a useful suggestion: "But utilitarianism, and most other forms of consequentialism, have highly implausible implications, which flow directly from the fact that their mode of justification is, at base, an aggregative one: the sum of a certain sort of value is to be maximized" [9] (p. 230). As long as a certain behavioral choice has certain positive effects, a moral justification can always be obtained through consequentialist principles. No matter how small the moral benefits of this choice are, the greatest good will always be provided to as many people as are involved. According to consequentialism, there is no doubt that we should give priority to this choice, but it turns out to be morally wrong in cases like Risky Electricity. Different from this, the Moral Relevance Approach does not regard the individual's moral claim as a limited and fixed-value carrier but as a normative precondition that cannot be calculated simply. Before we apply a consequentialist principle, whether it is morally relevant to the agent should be determined first. By doing this, rescuing an individual but sacrificing irrelevant interests could also be morally right, just as we found in Risky Electricity.

Lastly, the Moral Relevance Approach does not exclude the importance of integrated benefits in moral decision-making. What the Moral Relevance Approach provides is a precondition before we apply the consequentialist principles. The greatest good is still a worthy ethical target that we should pursue. Moral relevance is taken as an additional constraint on the consequentialist principle. There is no doubt that the greatest good is still the desirable ethical target that we should pursue. Since both consequentialism and nonconsequentialism cannot satisfactorily explain why we should save the driver instead of preventing a possible electricity failure in Risky Electricity, the Moral Relevance Approach is advocated for this type of moral situation. According to Seth Lazar, "Our duties of rescue and relevance are closely connected—and this connection allows us to resist aggregation without endorsing the contractualist moral theory that usually underpins it" [13] (p. 120). By introducing moral relevance as the decisive ethical normative factor, a reasonable moral theory could possibly meet the special requirements of AI.

What the Moral Relevance Approach supplies is a considerable normative foundation for AI ethics that can be used to decide which choice is morally right or morally wrong for human beings. That is the reason why it is ethically significant to AI. Compared with other normative theories, the Moral Relevance Approach could be regarded as a constraint to the consequentialist principle. In this way, the Moral Relevance Approach could explain why our moral intuitions and judgments contradict consequentialism in Risky Electricity.

## 5. Integrated Ethical Foundation

The upshot is that the Moral Relevance Approach provides us with a considerable ethical foundation for AI ethics. Generally speaking, it is the understandings of moral rightness that shape our ethical consensus and propel our moral motivations. The cognition of moral rightness constitutes our primary ethical reason and our behavioral motivation. Only when moral rightness is sufficiently acknowledged do ethical agents have satisfactory reasons to behave as this rightness suggests. As consequentialism implies, an action is morally wrong because it is not favorable in terms of combined benefits, while nonconsequentialism holds a different ethical normative understanding of this. Across the various contractualist constructions, their common claim is that combined benefits are not the only ethical normative principle. According to Kantian contractualism, an act is morally wrong because it violates some absolute moral order. Following nonconsequentialist interpretations, an action is morally wrong because it is rejected by the moral consensus extracted from contractualist procedures. Different from these approaches, the Moral Relevance Approach supplies a thought-provoking understanding of moral rightness and wrongness.

What the Moral Relevance Approach provides is a different moral foundation inspired by the works of T. M. Scanlon, in which an action is morally right because it cannot be reasonably rejected. According to Scanlon, "deciding whether an action is right or wrong requires a substantive judgment on our part about whether certain objections to possible moral principles would be reasonable" [9] (p. 194). This approach leads to a considerable means of taking others' interests into account. It is also known as the Great Burden Standard: "It would be unreasonable to reject a principle because it imposed a burden on you when every alternative principle would impose much greater burdens on others (at least one person)" [14] (p. 259). It does not merely rely on a common ethical target to obtain a moral consensus between different individuals.

During consequentialist and nonconsequentialist analyses of morally right and wrong decisions, the first and foremost is that they all presuppose some worthy targets in moral decision-making. As for consequentialism, the greatest good for the greatest number of people is what we should pursue; however, for nonconsequentialism, ethical aims are converted into nonutilitarian ethical aims such as absolute moral orders and reasonable moral capacity. No matter which ethical goals we hold, it is crucial that the ethical target must not be refused by other relevant people. In other words, it is the common ethical appeals that connect us with other people and provide other people with the reasons to accept their corresponding choices. This is where the obstacle of how others can be

persuaded to receive and execute such targets arises. It would be easy to balance the benefits and burdens when they all pertain to a single person. Just as we found in Risky Highway and Risky Electricity, this question becomes rather tricky when it applies to intrapersonal situations. That is to say, when the benefits pertain to some people and burdens to others, balancing them and calculating the greatest good becomes rather more difficult. "The difference between the unity of the individual and the separateness of persons requires that there be a shift in the moral weight that we accord to changes in utility when we move from making intrapersonal trade-offs to making interpersonal tradeoffs" [15] (p. 381). Most consequentialism and nonconsequentialism cannot satisfactorily account for the change in moral weight. What we hold in previous cases may turn out to be useless in the next case. Due to the distinctiveness of practical requirements and contradictory moral judgments, AI ethics thus appeal to a normative theory that could integrate our consequentialist and nonconsequential intuitive concerns.

According to the Moral Relevance Approach, an action is morally wrong because it is excluded by some set of moral normative principles that people cannot reasonably reject. This approach plays an important role in the explanation of moral rightness in Risky Electricity, as we saw. The importance of the Moral Relevance Approach is that it brings forth a different understanding of the relationship between moral normativity and an inability to justify. This is the common-sense approach to moral normativity. For an action to be morally wrong, it means that the action violates some crucial moral principle, thus making it morally wrong. According to Scanlon's definition, an inability to justify an action should be not the consequence of moral wrongness but the reason why it should be morally wrong. The redundancy objection is raised as follows: "Whenever principles allowing an action are reasonably rejectable because that action has some feature (or set of features) F, the action is wrong simply in virtue of being F and not because its being F makes principles allowing it reasonably rejectable" [16] (p. 337). If this objection is proven to be true, the Moral Relevance Approach and its contractualist formula will be significantly undermined.

What the redundancy objection declares is that the Moral Relevance Approach may cause problems of concealed logical tautology. According to the redundancy objection, reasonable rejection should, instead, be regarded as a redundancy description of the higher-level wrong-making property. As Philip Stratton-Lake puts it, "In doing this they show that Scanlon should abandon his view that wrongness is a reason-providing property" [17] (p. 76). It is the moral fact or other moral property that decides whether this action is morally right, while reasonable rejection non-causally leads to the action being morally wrong in a trivial sense. As Derek Parfit puts it, Scanlon's formula would then be another concealed tautology, one of whose open forms would be the claim that acts are disallowed by such unrejectable principles just when these acts are disallowed by such principles [18] (pp. 369–370). Reasonable rejection adds nothing to the explanation of moral rightness and, thus, should be abandoned.

The redundancy objection mistakes the reason-providing property as the main moral framework that the Moral Relevance Approach attempts to provide. Reasonable rejection is the substantive definition of moral rightness. A first-order definition of moral rightness is not the primary concern of the Moral Relevance Approach. As Scanlon puts it, "What is basic in my view is what no one could reasonably reject, not what certain people do or hypothetically would agree to (or not reject) under some specified conditions" [19] (p. 435). This distinction looks small, but it occupies a vital role in understanding the major ethical aim of the Moral Relevance Approach. Any substantive moral content will not be given directly. A hypothetical contractualist construction would be a futile attempt according to this view. Accepting that an action is morally wrong means that the moral agent will have sufficient and convincing moral reasons not to do it. This does not lead to the implication that moral wrongness will provide the moral agents with moral reasons not to behave correspondingly. The redundancy objection misunderstands this and, thus, becomes a futile attempt.

Drawing on an analogy with natural kind terms is not appropriate to clarify the implication of moral wrongness. Just like blueness is the common property that all blue things have, the redundancy objection also regards moral wrongness as a certain common property shared by many actions considered morally wrong. This analogy is inappropriate to describe the diverse definitions of moral wrongness. As Scanlon puts it, "when different people call actions morally wrong some of them may have in mind different standards, and different reasons that they take to support them" [19] (p. 438). What the Moral Relevance Approach provides is a feasible ethical foundation to connect us with other people, not a description of what a morally wrong property is. If we concentrate on the definition of moral rightness and wrongness, a latent conflict between consequentialism and nonconsequentialism will foreseeably emerge, just as we have seen in Risky Highway and Risky Electricity.

The so-called sympathetic identification constitutes our most moral motivations and behavioral attitudes. As Alex Voorhoeve puts it, "Psychological research suggests that, as a consequence, moral reasoning that relies solely on this process of sympathetic identification will not account for numbers" [20] (p. 69). This is also the reason why the utilitarian principle is not applicable to morally relevant cases like Risky Electricity. Compared with the benefits of strangers, prioritizing benefits to the self is permissible in normal moral situations.

Reasonable rejection thus becomes a potential way for AI ethics to integrate the two major properties of our moral ability, which are the separateness of the individual and the unity of collectivity. The former asks moral agents to show separate and equal concern toward each individual's benefits and burdens. "Even though we cannot contain all these separate lives together in our imagination, their separateness must be preserved somehow in the system of impersonal values which impartiality generates" [21] (p. 66). The unity of collectivity asks moral agents to take a more impersonal standpoint and chase the most worthy outcomes under the restraint of the Moral Relevance Approach. Using this approach, the moral agent does not make moral judgments from a third-person perspective, like consequentialism and nonconsequentialism do, but imaginatively places the moral agent in the first-person perspective of each person involved in moral decision-making. By doing so, AI ethics could integrate different individual moral claims without encountering the obstacles faced by consequentialism and nonconsequentialism. When each individual is considered separately, everyone has to compare their moral claim with others' competing claims and determine which is the strongest moral claim we should all try to circumvent.

## 6. Conclusions

Faced with the distinctiveness of AI ethics, this article offers the Moral Relevance Approach as a considerable normative foundation to satisfactorily integrate our consequentialist and nonconsequentialist moral judgments. Taking moral relevance as the individual constraint of consequentialist principles, the Moral Relevance Approach aims to provide a possible way to connect others with ourselves.

First, individual moral claims should not be excluded from the moral decision-making process. A purely ex post analysis of combined benefits cannot distinguish the moral choices accompanied by the same consequences. Morally relevant factors, especially the motivations and reasons of the moral agents who are influenced, could also generate a considerable impact on moral normativity.

Second, the reasonable rejection adopted by the Moral Relevance Approach is not a redundant explanation for moral rightness. It is essential for AI ethics to take reasonable rejection into account in moral decision-making. This is not only the common ethical target that shapes our moral consensus but also the inherent moral ability to connect with others.

**Funding:** This research was funded by National Office for Philosophy and Social Sciences of China [22BZX014].

**Institutional Review Board Statement:** Not applicable.

**Informed Consent Statement:** Not applicable.

**Data Availability Statement:** No new data were created or analyzed in this study. Data sharing is not applicable to this article.

**Conflicts of Interest:** The author declares no conflicts of interest.

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
