# Peer review of "Moral Relevance Approach for AI Ethics"

_philosophies, doi:10.3390/philosophies9020042_

Round 1

Reviewer 1 Report

Comments and Suggestions for Authors

The paper is obviously relevant because it addresses a matter that scientists are increasingly concerned about, namely the ethical difficulties raised by the use of artificial intelligence. The article in question is one of several attempts to meaningfully address these issues. And, while the paper delves deeply into consequentialist and non-consequentialist moral theories, it is unclear why it refers to a "morally relevant approach to AI ethics" as the only "morally relevant approach." More specifically, because the author or authors of the article believe their technique to be ethically significant, proponents of other moral theories will likewise view their approach as moral. Along with correcting some minor errors (line 321 misspelt Derek Parfit's name), it is advisable to consider revising the name of the defined approach.

Author Response

1. Summary

Thank you very much for taking the time to review this manuscript. It is appreciated to receive such useful advice. Please find the detailed responses below and the corresponding corrections highlighted in the re-submitted files.

2. Questions for General Evaluation

Reviewer’s Evaluation

Response and Revisions

Is the content succinctly described and contextualized with respect to previous and present theoretical background and empirical research (if applicable) on the topic?

Yes

corresponding response can be found in the point-by-point response.

Are all the cited references relevant to the research?

Yes

Are the research design, questions, hypotheses and methods clearly stated?

Can be improved

Are the arguments and discussion of findings coherent, balanced and compelling?

Can be improved

For empirical research, are the results clearly presented?

Is the article adequately referenced?

Not applicable

Yes

Are the conclusions thoroughly supported by the results presented in the article or referenced in secondary literature?

Yes

3. Comments and Suggestions for Authors:

Comments 1:
The paper is obviously relevant because it addresses a matter that scientists are increasingly concerned about, namely the ethical difficulties raised by the use of artificial intelligence. The article in question is one of several attempts to meaningfully address these issues. And, while the paper delves deeply into consequentialist and non-consequentialist moral theories, it is unclear why it refers to a "morally relevant approach to AI ethics" as the only "morally relevant approach." More specifically, because the author or authors of the article believe their technique to be ethically significant, proponents of other moral theories will likewise view their approach as moral. Along with correcting some minor errors (line 321 misspelt Derek Parfit's name), it is advisable to consider revising the name of the defined approach.

Response 1: Thank you for pointing this out. It is advisable that the connection between moral relevant approach and AI ethics should be clarified clearly in this paper. AI ethics focus on the ethical issues accompanied with the widespread application of AI technology, such as privacy leakage, discrimination, unemployment and morally wrong decision-making. The definition of AI ethics has been added in line 16-21.

“The widespread application of AI technology inevitably generates great impact on existing social order and human ethical cognition. AI ethics thus is proposed as an emerging and interdisciplinary field concerned with addressing ethical issues of AI. What AI ethics most concern is the ethical risks especially brought by the AI systems, such as privacy leakage, discrimination, unemployment and morally wrong decision-making.”

This paper focus on the issue of moral decision-making, which is also AI ethics most concern. Relevant research in this field has been introduced in line 23-38.

“Some AI systems try to translate certain moral normative principles into their algorithms. For instance, Derek Leben has presented a way of developing Rawls’ Contractarian moral theory into an algorithm for car crash optimization in autonomous vehicles [3] (p.114). According to this, AI could decide who should be saved at the first place and which decision is morally right by the given normative principles such as the Maximin principle. Compared with Utilitarainism, Kantian Ethics, Virtue Ethics, and Prima Facie approaches, Derek Leben thinks Rawlsian Contractarianism could behave better. Faced with the contradictory moral claims in particular moral situations, Derek Leben choses to satisfy the maximal benefits of worse off as the normative principle of AI ethics. The problem lies in that Rawlsian Contractarianism could not integrate our human consequentialist and nonconsequentialist satisfactorily. What Rawlsian Contractarianism tries to provide is a considerable ethical principle for moral decision-making. The major risk of AI moral decision-making is not only the uncertainty due to the opacity of technology, but also is its astonish rapid growth has excessed the boundaries of human ethical cognition. What hinders the AI ethics is that people hold different normative claims due to their diverse ethical concerns.”

“What the moral relevant approach supplies is a considerable normative foundation for AI ethics, which can be used to decide which choice is morally right or morally wrong to human beings. That is the reason why it is ethically significant to AI ethics.”

Thank you for pointing this out. The misspelling of Derek Parfit's name has been corrected.

4. Response to Comments on the Quality of English Language

Point: English language fine. No issues detected

Response: Thank you for the comment. It is appreciated to received such comment.

Reviewer 2 Report

Comments and Suggestions for Authors

General comments

Although the article addresses an interesting and current topic, it is somewhat awkward. It needs to be clearly defined what kind of ethics and morality we are discussing about humans and their interpersonal relations (entire chapters 4 and 5 are devoted to this) or the human-AI relationship, which the title promises us. In the introduction, I miss a more detailed analysis of the situation (AI ethics examples 135 mil. results, AI ethics Books - 130 mil. Results, AI ethics articles - 412 mil. results, AI ethics and Moral Relevant Approach - 81 mil. results) and what is in your to the contribution of a different or the new one. There is a lack of an overview of what is generally happening in the field of AI ethics (analysis of various models, literature review), a lack of general definitions of what you mean by AI ethics, which is not clear, or the human ethics of using AI, or the ethical norms that must be implemented in AI algorithms, or...

Particular comments

The abstract must be rewritten

Define what we are talking about human ethics and morals or ethics that AI must follow and must be "written" in the algorithm, e.g. in the form of Asim's Laws of Robotics?

 The rapid growth of AI technology has exceeded the boundaries of human ethical cognition. Unintelligible, whose ethics, human or the ethics that man is trying to prescribe for AI?

Taking moral relevance as the precondition of the consequentialist principles, the Moral Relevant Approach aims to consider individual moral claims plausibly.

 The AI ethics appeal to a concrete answer to moral decision-making. Who, humans using AI or AI itself?

1. Introduction

To satisfy the concrete practical requirements of AI ethics, a unique ethical foundation is provided by the Moral Relevant Approach to integrate our contradictory intuitive judgments. What mean our - human or AI??

2. Distinctiveness of AI ethics

You talk about AI ethics, which must be implemented in AI algorithms and their problems, for example classical MIT ethical problem.

Question: Here comes the question: which moral normative theory should this AI take? is wrong - what moral norms should be implemented in the AI of a self-driving vehicle? I repeat, be consistent, what ethics and morals are you talking about?

distinctiveness of AI ethics must be clarified. How??

77 Risky Highway, there is no time for AI to argue which ethical normative theory is better?? AI algorithms are optimization algorithms that are always looking for the best solution, they can calculate hundreds of options and choose the best one in a tenth of a second. The only question is, is this the best ethical solution at the given moment from a human point of view??

The AI ethics require a more concrete answer with specific practical requirements. I agree, but are we capable of creating them.

102-103 Both consequentialism and nonconsequential ism cannot be the normative moral base of our intuitive judgments in the autonomous vehicle case. 

AI is not capable of intuitive decision-making, but only (statistical, computational) optimization decision-making, based on the analysis of available data!!

3. Morally Relevant Approach

Risky Electricity: The AI of this vehicle has to make a moral decision/.../ saving the driver; or letting a nearby city get in an insignificant power failure/.../ - the case is not the most fortunate, you are comparing a "simple, one-dimensional" case with a complex/multidimensional one - what if it will die due to a power failure, e.g. 100 people in the hospital?

4. Reasons for the Morally Relevant Approach

For example: killing someone and letting someone die is morally equivalent per se, according to the consequentialist principle. 

How does this relate to AI ethics - not consistent?

265 the action is morally wrong is because - proof reading needed!

346 What the Moral Relevant Approach provides is a feasible ethical foundation to connect us with other people, not a description about what is the morally wrong property. 

How does this relate to AI??

367 the first-person perspective of each person who is involved in moral decision-making. Where is the AI here?

etc, etc.

Author Response

Response to Reviewer 2 Comments

1. Summary

Thank you very much for taking the time to review this manuscript. It is appreciated to receive such useful advice. The morals and ethics this paper talking about have been defined. Some relevant researches about AI ethics have been added. Every comment has received a point-to-point response. Please find the detailed responses below and the corresponding corrections highlighted in the re-submitted files.

2. Questions for General Evaluation

Reviewer’s Evaluation

Response and Revisions

Is the content succinctly described and contextualized with respect to previous and present theoretical background and empirical research (if applicable) on the topic?

Must be improved

corresponding response can be found in the point-by-point response.

Are all the cited references relevant to the research?

Must be improved

Are the research design, questions, hypotheses and methods clearly stated?

Can be improved

Are the arguments and discussion of findings coherent, balanced and compelling?

For empirical research, are the results clearly presented?

Is the article adequately referenced?

Are the conclusions thoroughly supported by the results presented in the article or referenced in secondary literature?

Must be improved

Not applicable

Must be improved

Must be improved

3. Comments and Suggestions for Authors:

General comments:

Although the article addresses an interesting and current topic, it is somewhat awkward. It needs to be clearly defined what kind of ethics and morality we are discussing about humans and their interpersonal relations (entire chapters 4 and 5 are devoted to this) or the human-AI relationship, which the title promises us. In the introduction, I miss a more detailed analysis of the situation (AI ethics examples 135 mil. results, AI ethics Books - 130 mil. Results, AI ethics articles - 412 mil. results, AI ethics and Moral Relevant Approach - 81 mil. results) and what is in your to the contribution of a different or the new one. There is a lack of an overview of what is generally happening in the field of AI ethics (analysis of various models, literature review), a lack of general definitions of what you mean by AI ethics, which is not clear, or the human ethics of using AI, or the ethical norms that must be implemented in AI algorithms, or...

Response: Thank you very much for taking the time to review this manuscript. It is appreciated to receive such useful advice. I entirely agree with the comment. AI ethics focus on the ethical issues accompanied with the widespread application of AI technology, such as privacy leakage, discrimination, unemployment and morally wrong decision-making. The definition of AI ethics has been added in line 17-21. What AI ethics most concern have been also introduced.

“AI ethics thus is proposed as an emerging and interdisciplinary field concerned with addressing ethical issues of AI. What AI ethics most concern is the ethical risks brought by the AI systems, such as privacy leakage, discrimination, unemployment and morally wrong decision-making.”

This paper focus on the issue of moral decision-making, which is also the issue AI ethics most concern. Some relevant researches in this field has been introduced in line 23-38.

“Some AI systems try to translate certain moral normative principles into their algorithms. For instance, Derek Leben has presented a way of developing Rawls’ Contractarian moral theory into an algorithm for car crash optimization in autonomous vehicles [3] (p.114). According to this, AI could decide who should be saved at the first place and which decision is morally right by the given normative principles such as the Maximin principle. Compared with Utilitarainism, Kantian Ethics, Virtue Ethics, and Prima Facie approaches, Derek Leben thinks Rawlsian Contractarianism could behave better. Faced with the contradictory moral claims in particular moral situations, Derek Leben choses to satisfy the maximal benefits of worse off as the normative principle of AI ethics. The problem lies in that Rawlsian Contractarianism could not integrate our human consequentialist and nonconsequentialist satisfactorily. What Rawlsian Contractarianism tries to provide is a considerable ethical principle for moral decision-making. The major risk of AI moral decision-making is not only the uncertainty due to the opacity of technology, but also is its astonish rapid growth has excessed the boundaries of human ethical cognition. What hinders the AI ethics is that people hold different normative claims due to their diverse ethical concerns.”

Comments 1:

The abstract must be rewritten

Define what we are talking about human ethics and morals or ethics that AI must follow and must be "written" in the algorithm, e.g. in the form of Asim's Laws of Robotics?

The rapid growth of AI technology has exceeded the boundaries of human ethical cognition. Unintelligible, whose ethics, human or the ethics that man is trying to prescribe for AI?

Taking moral relevance as the precondition of the consequentialist principles, the Moral Relevant Approach aims to consider individual moral claims plausibly.

The AI ethics appeal to a concrete answer to moral decision-making. Who, humans using AI or AI itself?

Response 1: Thank you for pointing this out. I agree with this comment. The abstract has been rewritten.

“Abstract: Artificial intelligence (AI) ethics is proposed as an emerging and interdisciplinary field concerned with addressing ethical issues of AI, such as the issue of moral decision-making. The conflict between our intuitive moral judgements constitutes the inevitable obstacle in moral decision-making of AI ethics. This article outlines the Moral Relevant Approach which could provide a considerable moral foundation for AI ethics. Taking moral relevance as the precondition of the consequentialist principles, the Moral Relevant Approach aims to consider individual moral claims plausibly. It is not only the common ethical target shaping our moral consensus, but also the inherent moral ability connecting others with us.”

The sentence in line 44-45 has also been corrected as below.

“A concrete normative principle is needed in moral decision-making of AI ethics.”

Comments 2: 1. Introduction

To satisfy the concrete practical requirements of AI ethics, a unique ethical foundation is provided by the Moral Relevant Approach to integrate our contradictory intuitive judgments. What mean our - human or AI??

Response 2: Thank you for pointing this out. The definition should be expressed exactly and precisely. What this paper refer is our human contradictory intuitive judgment. What this paper adopts is a normative approach, aimed to provide a corresponding and reasonable explanation of our different moral judgements. That means the decisions made by the AI should take human moral judgements as the behavioral criterions of AI. Accordingly, I have changed the statements to emphasize this point in line 54-59.

“To satisfy the concrete practical requirements of AI ethics, a unique ethical foundation is provided by the Moral Relevant Approach to integrate our human contradictory intuitive judgements. What the Moral Relevant Approach adopts could be taken as a normative approach, which aimed to provide a corresponding and reasonable explanation of our different moral judgements. That means the decisions made by the AI should be consistent with human moral judgements.”

Comments 3: 2. Distinctiveness of AI ethics

You talk about AI ethics, which must be implemented in AI algorithms and their problems, for example classical MIT ethical problem.

Question: Here comes the question: which moral normative theory should this AI take? is wrong - what moral norms should be implemented in the AI of a self-driving vehicle? I repeat, be consistent, what ethics and morals are you talking about?

distinctiveness of AI ethics must be clarified. How??

Response 3: Thank you for pointing this out. I completely agree with the comment. It is appreciated to received such useful comment. The sentence has been corrected as your advice in line 81-82.

“Here comes the question: what moral norms should be implemented in the AI of a self-driving vehicle?”

The morals and ethics what I’m talking about is the normative issue of AI ethics. More precisely, this paper aims to provide a normative principle for AI to make it be consistent with our human intuitive moral judgements.

Thank you for pointing this out. The sentence has been changed in line 83-84.

“To illuminate the issue of moral decision-making in AI ethics, there are two distinctiveness must be clarified as below.”

Comments 4: 77 Risky Highway, there is no time for AI to argue which ethical normative theory is better?? AI algorithms are optimization algorithms that are always looking for the best solution, they can calculate hundreds of options and choose the best one in a tenth of a second. The only question is, is this the best ethical solution at the given moment from a human point of view??

The AI ethics require a more concrete answer with specific practical requirements. I agree, but are we capable of creating them.

Response 4: Thank you for pointing this out. I agree with the comment. Therefore, I have rewritten the sentence in line 98-99.

“As we observed in Risky Highway, there is a unique requirement for AI to respond to the moral situations immediately.”

The same as the sentence in line 103-104. The point is that how would the AI make moral decisions, and how to make this process transparent to us.

“To make the AI be transparent to us, a concrete normative principle about how would the AI act is required.”

Comments 5: 102-103 Both consequentialism and nonconsequential ism cannot be the normative moral base of our intuitive judgments in the autonomous vehicle case. 

AI is not capable of intuitive decision-making, but only (statistical, computational) optimization decision-making, based on the analysis of available data!!

Response 5: Thank you for pointing this out. I agree with the comment. Therefore, I have changed expression in line 124-125.

“Both consequentialism and nonconsequentialism are faced with the same difficulty in AI ethics.”

Comments 6: 3. Morally Relevant Approach

Risky Electricity: The AI of this vehicle has to make a moral decision/.../ saving the driver; or letting a nearby city get in an insignificant power failure/.../ - the case is not the most fortunate, you are comparing a "simple, one-dimensional" case with a complex/multidimensional one - what if it will die due to a power failure, e.g. 100 people in the hospital?

Response 6: Thank you for pointing this out. I agree with the comment that this case is not the most fortunate. This case is proposed to illuminate the importance of moral relevance. Combined benefits aren’t the only standard of moral rightness. As long as we comparing with the seriousness of different moral choices before moral decision-making, its theoretical goal has been accomplished. Just as the Transmission case constructed by T.M. Scanlon shown, we owe to save the work’s arm for its moral seriousness. Likely, Risky Electricity is sufficient to justify the importance of moral relevance.

Comments 7: 4. Reasons for the Morally Relevant Approach

For example: killing someone and letting someone die is morally equivalent per se, according to the consequentialist principle.

How does this relate to AI ethics - not consistent?

Response 7: Thank you for pointing this out. I agree with the comment. Therefore, I have added some content to clarify the connection between this example and AI ethics in line 236-240.

“This instance is presented to illustrate that AI could not make the moral judgements if it merely relies on the consequentialist principle, since different behaviors with the same consequence have different moral evaluations due to their various motivations and intentions. Thus, it is essential to introduce the moral relevance into the moral decision-making of AI ethics.”

Comments 8: 265 the action is morally wrong is because -proof reading needed!

Response 8: Thank you for pointing this out. I agree with the comment. The wrong sentence has been corrected in line 287-288.

“As consequentialism implies, the action is morally wrong because it is not favorable for the combined benefits.”

Comments 9: 346 What the Moral Relevant Approach provides is a feasible ethical foundation to connect us with other people, not a description about what is the morally wrong property.

How does this relate to AI??

Response 9: Thank you for pointing this out. I agree with the comment. I have added some content to illustrate the connection between the Moral Relevant Approach and AI ethics in line 274-276.

“What the Moral Relevant Approach supplies is a considerable normative foundation for AI ethics, which can be used to decide which choice is morally right or morally wrong to human beings. That is the reason why it is ethically significant to AI ethics.”

Comments 10: 367 the first-person perspective of each person who is involved in moral decision-making. Where is the AI here?

Response 10: Thank you for pointing this out. I agree with the comment. The connection with AI ethics has not been fully explained. Therefore, some sentences have been added in line 396-398.

“By doing so, AI ethics could integrate different individual moral claims without en-countering the obstacles consequentialism and nonconsequentialism have been faced.”

4. Response to Comments on the Quality of English Language

Point: I am not qualified to assess the quality of English in this paper

Response: Thank you for the comment. It is appreciated to receive such useful advices. Minor English errors have been corrected and the whole manuscript has been polished.

Reviewer 3 Report

Comments and Suggestions for Authors

I have the impression that the article is well organized, but the arguments could be more emphatically documented. Probably, the reader needs more practical instances of the problems of ethical theory. Thus, the supported thesis for the applicability of the Moral Relevant Approach, seems too sophisticated and not so passionately supported. The improvement required may be applied to the instances of situations already mentioned by the author. I would appreciate some kind of subtle description of the risky circumstances, in a way that should prove the relevance of the position.   

The problem I detected can be elucidated by citing the following part of the article: "According to F. M. Kamm, killing someone and letting someone die have distinct ethical evaluations due to the difference of their moral agents’ diverse motivations and intentions". I do not regard as persuading and sincere the examination of that situation above, since it is not adequately analysed in its essential depth and perspectives.

Comments on the Quality of English Language

I find the level of English language advanced.

Author Response

Response to Reviewer 3 Comments

1. Summary

Thank you very much for taking the time to review this manuscript. It is appreciated to receive such useful advice. Please find the detailed responses below and the corresponding revisions/corrections highlighted/in track changes in the re-submitted files.

2. Questions for General Evaluation

Reviewer’s Evaluation

Response and Revisions

Is the content succinctly described and contextualized with respect to previous and present theoretical background and empirical research (if applicable) on the topic?

Can be improved

corresponding response can be found in the point-by-point response.

Are all the cited references relevant to the research?

Yes

Are the research design, questions, hypotheses and methods clearly stated?

Can be improved

Are the arguments and discussion of findings coherent, balanced and compelling?

Must be improved

For empirical research, are the results clearly presented?

Is the article adequately referenced?

Not applicable

Can be improved

Are the conclusions thoroughly supported by the results presented in the article or referenced in secondary literature?

Must be improved

3. Comments and Suggestions for Authors:

Comments :
I have the impression that the article is well organized, but the arguments could be more emphatically documented. Probably, the reader needs more practical instances of the problems of ethical theory. Thus, the supported thesis for the applicability of the Moral Relevant Approach, seems too sophisticated and not so passionately supported. The improvement required may be applied to the instances of situations already mentioned by the author. I would appreciate some kind of subtle description of the risky circumstances, in a way that should prove the relevance of the position.

The problem I detected can be elucidated by citing the following part of the article: "According to F. M. Kamm, killing someone and letting someone die have distinct ethical evaluations due to the difference of their moral agents’ diverse motivations and intentions". I do not regard as persuading and sincere the examination of that situation above, since it is not adequately analysed in its essential depth and perspectives.

Response 1: Thank you for pointing this out. I agree with the comment. More rigorous arguments should be given. Therefore, I have introduced a relevant research to the issue of moral decision-making in AI ethics in line 23-38.

“Some AI systems try to translate certain moral normative principles into their algorithms. For instance, Derek Leben has presented a way of developing Rawls’ Contractarian moral theory into an algorithm for car crash optimization in autonomous vehicles [3] (p.114). According to this, AI could decide who should be saved at the first place and which decision is morally right by the given normative principles such as the Maximin principle. Compared with Utilitarainism, Kantian Ethics, Virtue Ethics, and Prima Facie approaches, Derek Leben thinks Rawlsian Contractarianism could behave better. Faced with the contradictory moral claims in particular moral situations, Derek Leben choses to satisfy the maximal benefits of worse off as the normative principle of AI ethics. The problem lies in that Rawlsian Contractarianism could not integrate our human consequentialist and nonconsequentialist satisfactorily. What Rawlsian Contractarianism tries to provide is a considerable ethical principle for moral decision-making. The major risk of AI moral decision-making is not only the uncertainty due to the opacity of technology, but also is its astonish rapid growth has excessed the boundaries of human ethical cognition. What hinders the AI ethics is that people hold different normative claims due to their diverse ethical concerns.”

It is advisable that some instances should be furtherly demonstrated. The arguments should get more analysis from multiple perspectives to be more convincing. As it for the example of “killing someone and letting someone die”, I have added some content to clarify the connection between this example and AI ethics in line 236-240.

“This instance is presented to illustrate that AI could not make the moral judgements if it merely relies on the consequentialist principle, since different behaviors with the same consequence have different moral evaluations due to their various motivations and intentions. Thus, it is essential to introduce the moral relevance into the moral decision-making of AI ethics.”

More analyses have been added to highlighted the connection between this paper and AI ethics.

4. Response to Comments on the Quality of English Language

Point: Minor editing of English language required. I find the level of English language advanced.

Response: Thank you for pointing this out. It is appreciated to receive such useful comments. Minor English errors have been corrected and the whole manuscript has been polished.

Round 2

Reviewer 1 Report

Comments and Suggestions for Authors

The theses of the paper are clearer after being supplemented. Thank you!

Author Response

Thank you for the comments! This paper benefits enormously from your effort.

Reviewer 2 Report

Comments and Suggestions for Authors

The article is significantly improved and easier to read. Of course, much more could be done, but the work needs to be finished somewhere.

Author Response

Comments 1: Is the content succinctly described and contextualized with respect to previous and present theoretical background and empirical research (if applicable) on the topic? Can be improved.

Response 1: Thank you for pointing this out. It is appreciated to receive the comment. Therefore, I have added some sentences as supplements in line 187-192.

“It gives more consideration to the motivations and reasons people ex ante hold in moral decision-making. According to Scanlon: “This might be justified on an ex ante basis because, if the number of beneficiaries is sufficiently great, and if we have no reason ex ante to believe that one is more likely to be the victim in a situation of this kind than to be a beneficiary, then one has no reason, ex ante to object to this principle” [11] (p.510).”

Comments 2: Are all the cited references relevant to the research? Can be improved.

Response 2: Thank you for pointing this out. It is appreciated to receive the comment. Therefore, I have added some relevant statements to further illuminate the connection between this paper and the references in line 148-150, and line 358-359.

“The problem lies in that sticking the pursuit of the combined benefits will result in some counterintuitive moral judgements we try to avoid.”

“A first-order definition about moral rightness is not the primary concern of the Moral Relevant Approach.”

Comments 3: Are the arguments and discussion of findings coherent, balanced and compelling? Can be improved.

Response 3: Thank you for pointing this out. It is appreciated to receive the comment. Therefore, some vague words have been also replaced in line 244, line 256 and line 264.

Besides, some expressions of moral rightness and moral wrongness have been modified.